

# Attention-aware with stacked embedding for sentiment analysis of student feedback through deep learning techniques

Shanza Zafar Malik[1], Khalid Iqbal[1], Muhammad Sharif[1], Yaser Ali Shah[1], Amaad Khalil[2], M. Abeer Irfan[2] and Joanna Rosak-Szyrocka[3]

[1] Department of Computer Science, COMSATS University Islamabad, Attock Campus, Attock, Punjab, Pakistan
[2] Department of Computer Systems Engineering, University of Engineering and Technology Peshawar, Peshawar, KPK, Pakistan
[3] Faculty of Management, Czestochowa University of Technology, Częstochowa, Poland

## ABSTRACT

Automatic polarity prediction is a challenging assessment issue. Even though polarity assessment is a critical topic with many existing applications, it is probably not an easy challenge and faces several difficulties in natural language processing (NLP). Public polling data can give useful information, and polarity assessment or classification of comments on Twitter and Facebook may be an effective approach for gaining a better understanding of user sentiments. Text embedding techniques and models related to the artificial intelligence field and sub-fields with differing and almost accurate parameters are among the approaches available for assessing student comments. Existing state-of-the-art methodologies for sentiment analysis to analyze student responses were discussed in this study endeavor. An innovative hybrid model is proposed that uses ensemble learning-based text embedding, a multi-head attention mechanism, and a combination of deep learning classifiers. The proposed model outperforms the existing state-of-the-art deep learning-based techniques. The proposed model achieves 95% accuracy, 97% recall, having a precision of 95% with an F1-score of 96% demonstrating its effectiveness in sentiment analysis of student feedback.

## INTRODUCTION

The best way to obtain information about the opinions, emotions, and reactions of students regarding their educational institutions would be through student feedback and its sentiment analysis, which is an approach to understanding the rationality of reviewers. Various kinds of student feedback include positive, negative, and unbiased or neutral. There are several methods for gathering student feedback, including surveys that might be both open-ended and closed-ended, depending on the requirements of the information collection process. Open-ended form is a technique in which students must adequately

Corresponding author
M. Abeer Irfan,
abeer.irfan@uetpeshawar.edu.pk

explain their arguments and cannot just answer yes or no to any question. Through this approach, it is possible to examine what the students want to say or their true sentiments, but it takes time. When using a close-ended method, students can simply choose from a small number of alternatives for information, like Boolean values or rating scales. This method offers little information; however, analyzing and computing the student feedback is simple and noise-free when compared to other strategies. Another method for gathering student input is through a small group instructional diagnosis, which is carried out by a professional who interacts directly with the students and asks about their comments, although the process' implementation is more expensive than that of other methods. Another strategy for gathering student feedback involves the use of Likert Scale questions, which emphasize areas that need improvement by using a 5- or 7-point scale to evaluate the appointed criteria. The major objectives of education are to enhance student learning and effectively transfer information to students for optimal results. Transparency, consultation, passion, and adaptation are the four fundamental factors that the effective teaching method considers. The productivity and efficiency of a teacher depend on their domain expertise and information delivery skills, and we may enhance the teaching-learning process by using student feedback. Students' involvement in the form of their comments can bridge the gap between class practitioners and improve the learning environment (*Rezaeinia et al., 2019*). Sentiment assessment of scholar feedback has the ability to enhance the teaching procedure by providing teachers with important insights that they can use to calibrate their method of teaching. Through this feedback, teachers can gain a deeper understanding of their students, which can lead to increased student engagement and participation (*Dos Santos & Gatti, 2014*). For introducing new policies to develop the educational sector, accurate and systematic sentiment analysis of student feedback might be important (*Yang & Chen, 2017*). Sentiment analysis of student feedback can be used to evaluate various aspects of an institution, such as student, teacher, and course perspectives, as well as facilities and curriculum. Different studies have employed various techniques to analyze student feedback sentiment, including machine learning (ML) classifiers such as Random Forest, decision trees, and lexicon-based algorithms. However, these techniques may not take into account the correlation between words in large sets of data. Multiple types of text embedding techniques have also been used for more effective polarity assessment of scholar's responses. Despite the use of deep learning (DL) classifiers like long short-term memory (LSTM), limitations still exist based on the parameters and embedding methods used.

The novelty and contributions of our research are as follows:

• This article proposes a hybrid model that uses ensemble learning-based text embedding, a multi-head attention mechanism, and a combination of deep learning classifiers to perform polarity classification on the student's response.

• Our proposed model addresses issues with word morphology and polysemy by utilizing techniques such as lemmatization and stemming to treat inflected forms of words as a single entity, while also considering their semantics.

• Our model effectively captures the contextualization and local context of student feedback by utilizing a multi-head attention mechanism.

• Our multi-layer hybrid model, which combines text embedding techniques, multi-head attention mechanism, and deep learning-based models such as Bi-LSTM, GRU, and BiGRU, which accurately categorized our textual data and determined the emotion level of comments *i.e.,* positive, negative, or neutral, as well as the themes of remarks *i.e.,* lecturer, training programs, facilities or others.

• Using our multi-layer hybrid model, more accurate results in the classification of textual data of student feedback were achieved as compared to other methods.

The rest of the article is as follows: 'Literature Review' provides the background for the research work. 'Materials & Methods' presents the methodology of the proposed technique. 'Results' discusses the results and 'Conclusions' concludes the work and provides insights into the future work.

## LITERATURE REVIEW

In recent studies, scientists are deploying and developing new state-of-the-art DL classifiers (*Aslam, Sargano & Habib, 2023*; *D'Antoni et al., 2021*; *Kufel et al., 2023*; *Kumar et al., 2023*). These new architectures are helping in categorizing or classifying various domains. These classifiers can be effectively deployed or trained for polarity assessment or prediction. In literature, DBN used in conjunction with mathematical representation of different words to predict government affairs stated in the blog posts or articles written in the Korean language, which used a dataset of 50,000 political articles (*Song, Park & Shin, 2019*). The findings showed improved outcomes and an accuracy of 81.8% in properly identifying labels. An attention based deep learning model consisting of bidirectional convolutional neural network-recurrent neural network (CNN-RNN) for polarity assessment for different datasets of short lengths which include textual responses from twitter and review sites (*Peng, Zhang & Liu, 2022*). They used glove embedding with attention based bidirectional CNN-RNN and achieved F1-score of 84% but their research does not consider different aspects of text in sentiment analysis. Techniques such as CNN, RNN, and deep neural network (DNN) have been utilized for polarity prediction (*Tang & Zhang, 2018*). ViHSD dataset has been used for detecting hate speech in Vietnamese language (*Luu, Nguyen & Nguyen, 2021*). Different deep learning techniques and transfer models for achieving higher accuracy were used. They utilized deep learning techniques of fastText along with CNN and fastText along with gated recurrent unit (GRU). They also used transfer models BERT, XLMR and Distil BERT for accurate hate speech detection and achieved highest accuracy of 86.88% by using BERT technique but their work does not cover the normalization of lexicon-based features and acronyms.

Logistic regression, Random Forest, and support vector machine (SVM) for detection on the UIT-ViCTSD dataset in the Vietnamese language were employed (*Nguyen, Nguyen & Nguyen, 2021*). They also used BiGRU, CNN, LSTM + PhoW2V, LSTM + fastText, but the highest accuracy of 80% for detecting constructiveness of speech was achieved using the combination of LSTM with fastText. However, logistic regression showed the highest accuracy of 90.27% for detecting toxicity in speech. The error rate is high due to the imbalance of classes in the dataset. A supervised learning technique to examine student

feedback datasets for sentiment analysis was employed (*Nguyen et al., 2018*). Naive Bayes, Maximum Entropy, LSTM, and Bi-LSTM were the algorithms utilized.

Following comparison, it was discovered that Bi-LSTM performed the best in both sentiment (F1-score: 92.0%) and topic classification (F1-score: 89.6%). A feedback analysis program was also created to give administrators a snapshot of student interests. The technology enables result synthesis and analysis, saving time on manual evaluations and giving useful information for enhancing teaching and learning quality.

A re-sampling technique of SMOTE for polarity assessment of scholar's response with different ML algorithms of SVM, Multinomial Naive Bayes (MNB), KNN, NN and Random Forest (RF) was suggested (*Flores et al., 2018*). They used a small Kaggle dataset of student feedback and achieved the accuracy of 84% but due to their small dataset, biasing factor is involved for achieving accuracy. For sentiment analysis, a brand-new deep model called ABCDM that outperforms previous models in terms of classification accuracy (*Basiri et al., 2021*). In order to understand both the past and the future context of the input text, ABCDM employs bidirectional LSTM and GRU networks along with pre-trained GloVe word embeddings. To make the semantic representations more informative, an attention layer is used to give particular words in a comment more weight.

The model achieved accuracy of 0.82 on sentiment140 dataset. In literature, a word book methodology for polarity classification of student feedback, using a dataset collected from students at their own educational institution, was employed (*Nasim, Rajput & Haider, 2017*). They used the TF-IDF technique with Lexicon-based features and achieved an accuracy of 93%. However, their research is limited to a specific domain and does not consider aspect-based sentiment analysis of student feedback. A solution to the problem of assessing teachers in education, based on student's input is presented in *Mabunda, Jadhav & Ajoodha (2021)*. It creates a sentiment analysis model that uses machine learning methods like SVM, MNB, Random Forests, K-nearest neighbor (K-NN), and neural networks to identify student input as positive, negative, or neutral. The model was trained using Kaggle student feedback data using feature engineering and re-sampling approaches. The findings indicated that the K-NN model was the most efficient in predicting student sentiment before resampling, with an accuracy of 81%, while the Neural Networks performed better after re-sampling, with an accuracy of 84%. This approach will assist educational institutions in making informed judgements regarding teaching and learning practices.

UIT-VSFC dataset in Vietnamese language for sentimental analysis of student feedback was used (*Nguyen, Van Nguyen & Nguyen, 2018*). They used ensemble architecture LSTM-DT on the data set and achieved F1-score of 90.2% for their deep learning model. However, their proposed model is not suitable for short sentences. In another study, the researchers used fuzzy-based approach for sentiment analysis of student feedback by using Data S1 of student feedback (*Asghar et al., 2020*). This fuzzy-based approach is based on different opinions of a single word and the polarity shifters. They achieved accuracy of 89% but their research work does not tackle the out-of-vocabulary issues. Sentiment analysis is an important topic in research; therefore, numerous researchers work on it in different domains. The short text representation has been focused for prediction of sentiments by using the IMDB dataset and the 20 Newsgroups dataset (*Liu et al., 2022*). They used

different text embedding techniques of CBOW, TF-IWF and LDA along with different machine learning techniques of KNN and SVM and achieved accuracy of 97% and 75% by applying dynamic fusion model with SVM on IMDB dataset. However, this research only focused on short text representation and does not consider end-to-end text representation.

Text classification using different Vietnamese datasets UIT-VSMEC, UIT-VSFC, HSD-VLSP including polarity prediction of feedback in academics by deploying deep learning architectures was used by *Huynh et al. (2020)*. They applied different deep learning and text embedding methods. They used Bi-LSTM with fastText, Bi-LSTM with Word2Vec, LSTM with fastText, LSTM with Word2Vec, GRU with fastText, GRU with Word2Vec, CNN with fastText, CNN with Word2Vec, BERT and achieved higher accuracy of 92.79% by using the ensemble method of GRU + CNN + BiLSTM + LSTM. In spite of that, their research work does not consider out-of-vocabulary issues. *Van Nguyen et al. (2018)* created a publicly available annotated corpus for Vietnamese sentence-level sentiment analysis. It included almost 16,000 annotated phrases with 91% agreement on feelings and 71% agreement on themes.

Using Maximum Entropy, the study produced the best overall F1-score of 88% for emotion polarity and 84% or higher for four separate themes. Sentiment analysis has grown in popularity on micro-blogging sites like Twitter in recent years, owing to the ease with which users share their ideas and thoughts. Previous research, however, has discovered limits in reliably assessing sentiment based on user context and confusing emotional information. *Bello, Ng & Leung (2023)* suggest employing BERT, along with other versions, for text categorization in natural language processing. An architecture which embodies the habit of the user from a document, to perform sentiment classification of tweets is presented. The proposed model was shown to be superior to typical vanilla baseline methods. It emphasized the importance of considering factors beyond a document's content in sentiment classification (*Alharbi & De Doncker, 2019*). The effectiveness of feedback is described in terms of its specific characteristics and external factors, such as the timing of input and the implications of both constructive and negative feedback (*Narciss, 2008*).

The BERT-DCNN model, employing BERT to generate word embeddings and three layers of DCNN to fine-tune the sentiment analysis model in smart city applications (*Jain et al., 2022*). CNNs are used for different purposes in sentiment analysis. A perceptible polarity analyzer has been proposed that predicts polarities content from the image (*Islam & Zhang, 2016*). In this research 1,269 images from twitter have been used and the results showed that Google-Net gave 9 percent more accurate performance than Alex-Net. The enhanced feature extraction was made possible by transforming Google Neural Network into a visual sentiment analysis framework. Hyper parameters were used to create a stable and dependable state. Deep learning approach for analyzing sentiment on Twitter data was used in literature as well (*Severyn & Moschitti, 2015*). Their contribution is in custom weights initialization in CNN. Training the model accurately without the need for additional features is crucial. They used an already trained twitter corpus in their word embedding layer. The model was able to predict polarity and achieved accuracy in message-level and phrase level tasks.

The RNN has a pre-defined tree structure and each node can have a unique matrix. RNN does not require input reconstruction. To overcome the limitations of existing models that rely on a large labeled corpus, a Tree-bank of Chinese social data sentiments was developed (*Li et al., 2014*). The RNN model predicted sentence-level polarities and it outperformed other models like SVM, Naive Bayes, and decision tress, *etc.*

The research used websites as a dataset that had different reviews on movies, and achieved higher accuracy than baseline models. The composition effects of phrases at different levels, by using an ensemble model that combines RNTN and polarity Tree-bank (TB) (*Socher et al., 2013*). To improve the accuracy of sentiment detection, more powerful composition models are needed, as current models using semantic word spaces are not effective in conveying the meaning of long phrases. This also requires more supervised resources for assessment and training. The RNTN model achieved approximately 80% on the predicted polarity of the sentences.

The RNN is a powerful classifier in natural language processing or for sequence data. It can capture context beyond written words, unlike traditional models that only consider the immediate history. A hierarchical bidirectional classifier for polarity assessment of customer response was proposed (*Silhavy et al., 2016*). The researchers investigated the feasibility of using a standard LSTM network to predict the polarity of online messages obtained from several social media sites (*Hassan et al., 2016*). The use of a simple LSTM network in polarity prediction yielded encouraging results, showing it to be a useful tool in the study of massive volumes of online data. The use of WSDNNs for Cross-Lingual Sentiment Classification by leveraging an Amazon dataset of reviews in four languages, each comprising 1,000 negative and 1,000 positive ratings was presented (*Zhou et al., 2016*). After experimentation on eighteen types of mixed language polarity assessment tasks, the authors found that the suggested strategy was more effective and successful than earlier techniques.

BERT encoder approaches have recently achieved considerable acceptance in the field of aspect-term sentiment analysis (ATSA). Typical techniques entail encoding the text and the aspect term separately, resulting in context and aspect word vectors. These initial vectors, however, lack semantic meaning and are easily altered by irrelevant words. To overcome these constraints, the CGBN model is introduced, which solely employs sentence sequences as input to the BERT encoder, allowing for the simultaneous extraction of context-hiding and aspect word-hiding vectors with rich semantic association information (*Peng, Xiao & Yuan, 2022*). The CGBN model also includes a new interactive gating mechanism called the co-gate, which effectively lowers the influence of noisy words and more effectively combines context and aspect term information, capturing emotional semantic aspects and reaching approximately 86% accuracy on twitter and restaurant reviews.

In related research, *Ruangkanokmas, Achalakul & Akkarajitsakul (2016)* suggest DBNFS, a modified variant of DBNs targeted at tackling the sentiment classification problem. The conventionally computationally intensive hidden layers in DBNs have been replaced by a filter-based feature selection approach based on chi-squared testing. This approach eliminates irrelevant characteristics while retaining the crucial ones, resulting in a more efficient and effective solution.

English is widely considered as an easily comprehensible language, however, there exists a variation known as "Roman Urdu" that resembles English but is not exactly the same. A multi-headed Bi-LSTM architecture designed for predicting the polarity of text written in Roman Urdu (*Chandio et al., 2022*).

The two datasets, RUECD and RUSA-19, which contained text written in Roman Urdu were used. The multi-headed Bi-LSTM architecture showed promising results (accuracy of 67%) in accurately determining the sentiment expressed in the text written in Roman Urdu.

A model that utilizes rhetorical structure theory to parse text, was proposed that fully employs the structural properties of LSTM to automatically enhance the core information and filter out peripheral information of text by establishing an LSTM network using the RST parse structure (*Fu et al., 2016*).

Furthermore, by representing the relationships between text segments, this approach improves text semantic representations. A capsule network model with Bi-LSTM was proposed (*Dong et al., 2020*). The experimental findings were presented on various datasets including MR, IMDB, and SST.

A hierarchical neural network that employs a hierarchical LSTM model was presented (*Chen et al., 2016*). This strategy is intended to concentrate on different semantic levels, allowing the model to effectively capture crucial semantic features. The LSTM model's hierarchical structure allows it to better comprehend the semantic linkages between words, resulting in enhanced performance in sentiment analysis tasks. The hierarchical neural network effectively captures the subtleties of language and properly predicts the sentiment portrayed in the text by paying attention to many semantic levels. To assess text polarity by extracting contextual information at the word level, the word context information and document representation can be used. This technique gives a more complete comprehension of the sentiment indicated in the text, resulting in better performance in polarity analysis tasks (*Han, Bai & Li, 2019*).

CNNs and RNNs were employed to develop an attention-based phrase assessment model (*Wang, Jiang & Yang, 2017*).

The results demonstrated the model's superior performance and efficacy in accurately assessing the sentiment reflected in the text. Multiple deep architectures can be jointly trained on multiple related tasks. The performance of a task can be improved by utilizing information from other related tasks (*Liu, Qiu & Huang, 2016*). The authors of a comparable work trained a model to predict the polarity of sentence-level content in both English and traditional Chinese (*Fu et al., 2017*). The model is a flexible tool for sentiment analysis jobs making it an important contribution to the area of sentiment analysis and achieving an average accuracy of 83.9%.

A dictionary infused LSTM classifier can be used that generates polarity embeddings for all words, including those not present in the lexicon (*Fu et al., 2018*). GRU has also been employed in literature quite efficiently. A GRU was used with polarity divided relations to model the influence of sentiment modifier context on sentimental relations in texts (*Chen, Zhuo & Ren, 2019*).

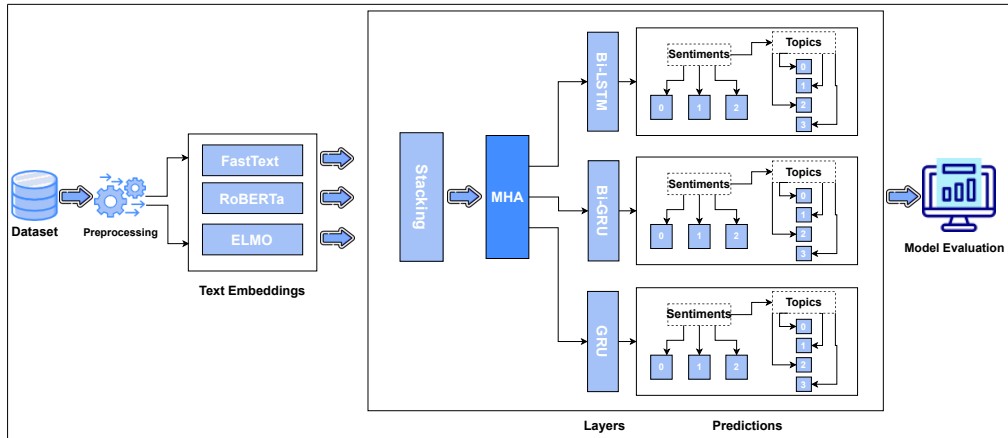

**Figure 1** **Flow diagram of proposed methodology** (*Imrana et al., 2021*; *Chung et al., 2014*; *Yu, Wang & Jiang, 2021*).

On the basis of the literature review, there is still a gap to improve the performance of the polarity classification. The proposed model which will be discussed in detail in the following 'Materials & Methods', improved: (1) The data representation, (2) intelligently selection and reassembling of the important features, (3) the weighting mechanism, which considers the sentimental relations in the texts, and (4) fine tuning of the deep learning classifiers. The experiments concludes that the proposed classifier not only captures sentimental relations but also excel vanilla baselines benchmarks.

# MATERIALS & METHODS

This section provides the methodology of our proposed work. Our anticipated model employs pre-processing, text embedding and a combination of multi-head attention with deep learning models to perform polarity classification on student's input. The research strategy is quantitative. Figure 1 depicts the proposed methodology.

## Pre-processing

In this section of the proposed model the unwanted artifacts were removed from the data (Vietnamese student feedback dataset). The dataset is polluted, with strange acronyms, spelling errors, emoticons or emojis, iconography, repeated letters, unwanted symbols, and numbers. These unwanted components were removed from the raw data by applying the following two step process:

### *Lemmatizer*

Lemmatization is a process that is used to group together the different inflected forms of a word in order to treat them as a single entity. This process is similar to stemming, which involves reducing words to their base form, but lemmatization also takes into consideration the semantics of the words. By grouping together words with similar meanings, lemmatization makes it easier to interpret the meaning of text by putting phrases together that have similar meanings (*Quan & Hung, 2021*).

### Tokenizer

Tokenization divides raw textual data into small, manageable pieces of standalone words called tokens. These tokens can be individual words or phrases, and are used to improve the interpretation of the text by providing context. Through tokenization, the original text is broken down into smaller segments that can be more easily analyzed and understood (*Singh & Singh, 2019*).

## Text embedding techniques

The proposed model uses an ensemble technique of text embedding, including static embeddings (such as Word2Vec and GloVe) and dynamic contextual embeddings (like BERT). These techniques are embedded to capture more linguistic features to effectively represent the syntax and semantics of student feedback. In our proposed model we used following text embedding techniques to convert the data in vector form:

### Fasttext

Fasttext embedding method is an improved version of word2vec that represents n-grams of characters for each word. This method resolves the morphological issue, comprehends the prefixes and suffixes of the provided text, and captures the meaning of short texts. While training a model, this strategy is quite helpful for resolving vocabulary gaps or missing words (*Reddy,* ). For example, in the given sentence "The curriculum is too small, and the content is not much different." the fastText will represent the word "small" like <<sm, sma, mal, all, al >> if $n = 3$.

### RoBERTa

A Robustly Optimized BERT Pre-training Approach (Roberta) is a robust optimized technique that is advanced version of BERT model. It is a transformer-based model based on self-attention mechanism that generates context for text in sentences. It is more suitable for large data by using efficient approach. During training it uses dynamic masking approach for specific and strong representation of text (*Liu et al., 2019*).

### Embeddings from Language Models

Embeddings from language models (ELMO) embedding approach may produce several embeddings for the same word used in various sentences in various contexts, making it particularly effective for capturing the deep context of words in sentences. The polysemous word problem, in which words used in various phrases might have several meanings depending on their context, can be handled with ELMo. ELMO employs a bidirectional language model to discover the deep context of the text (*Peng, Yan & Lu, 2019*). For example, in the given sentence: "The curriculum is too small, and the content is not much different." "We had difficulty with the small paper size, the model was large and complicated." The word "small" is used in different context so ELMO will generate different vector representation of this same word according to its context in different sentences that are given below: embedding for sentence 1: [−0.2566318 0.13504611 0.15567563 ... 0.21819718 0.23814695 −0.2749974]. Embedding for sentence 2: [−0.41841647−0.5392714 −0.38226548 ... 0.066176 0.21173294 −0.23470911]. ELMO

generate different vector representation for word "small" according to its context in both sentences.

## Stacking: ensemble encoded features

In this proposed module, three different types of features vectors are reassembled and more informative features are selected intelligently using the stacking method. The stacking ensemble approach, like the bagging ensemble mechanism for training multiple models, includes the creation of bootstrapped data subsets (*Pavlyshenko, 2018*). Here, in our case we have three types of feature subsets. These features are collectively fed into a classifier, known as a meta-classifier, which predicts the more suitable weighted feature vector. Meta-classifier has two layers architecture, to assess the input features and predict the output sampled features, correctly.

## Multi-head attention mechanism

This mechanism is used to enhance the model ability to capture contextual patterns and relevance of each word within the text. This layer in the proposed model helps in rearranging or identification of the translated word vectors in a succession of information that are crucial for determining the sentiment. Weights are allocated to various words in the context to determine their relative importance. In our proposed architecture, this layer is applied as a parallel attention strategy repeatedly to collect different aspects of the input data (*Niu, Zhong & Yu, 2021*).

## Deep learning classification models

Deep learning is the most advanced and latest technique used now a days for performing multiple tasks in different fields like sentiment analysis, image classification, artificial intelligence and many others. Deep learning layers were used in our suggested model. In order to handle to capture the temporal dependencies in the sequential textual data (student feedback), the proposed approach integrates several deep learning classifiers, like bidirectional long short-term memory (Bi-LSTM), gated recurrent units (GRU), and bidirectional GRU (Bi-GRU). These classifiers have more significance to classify sentiments in the textual data. To replicate the sequential dependencies between text in both forward and backward directions, as well as for text classification, we employed a Bi-LSTM layer, a multi head attention layer from the transformer's architecture. The outcome of multi-head attention was also fed into GRU and Bi-GRU, which accurately categorized our textual data and determined the emotion level of comments *i.e.,* positive, negative, or neutral, as well as the themes of remarks *i.e.,* lecturer, training programs, facilities or others.

### Bi-LSTM

The Bi-LSTM layer or architecture is mainly beneficial to use for sequence data. This processing model utilizes two LSTMs, one of which processes input forward, and the other in reverse. By using this structure, the network has access to additional data, which improves its understanding of context (*Imrana et al., 2021*). The architecture of Bi-LSTM is shown in Fig. 2.

LSTM is a famous RNN that can learn and recall information over long periods of time. LSTMs are particularly useful for natural language processing (NLP) and SR, where

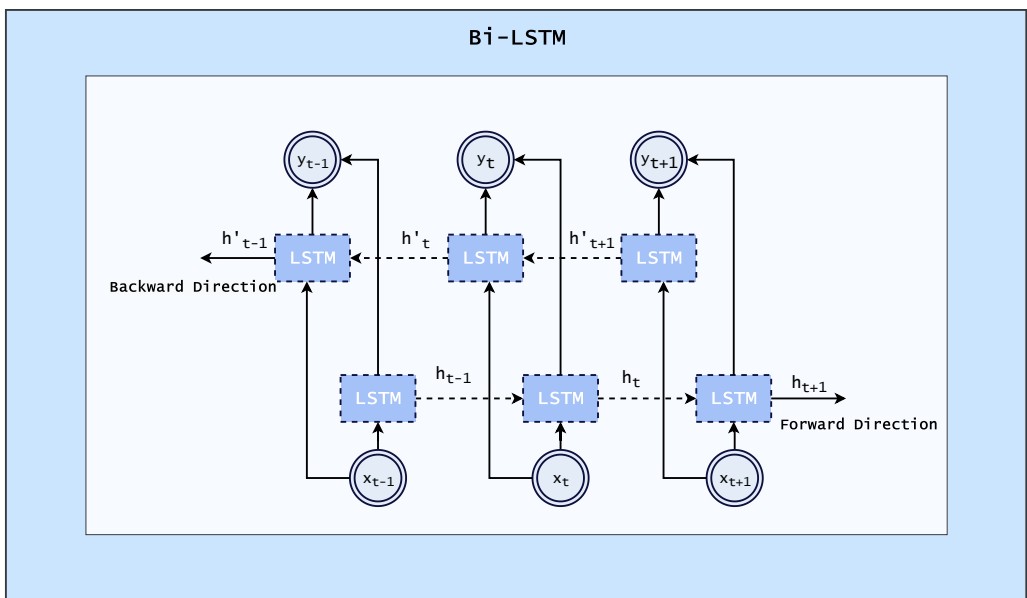

**Figure 2** **Architecture of Bi-LSTM (*Imrana et al., 2021*).**

input sequences can be fairly long and the ability to store information from earlier in the sequence is crucial. The LSTM architecture is made up of memory cells and three gate *i.e.,* input gates, forget gates, and output gates that allow it to selectively write to, read from, and erase its internal memory, allowing it to recall and use information from earlier in the input sequence. The architecture of LSTM is shown in Fig. 3.

### GRU

Gated recurrent unit (GRU) is another type of RNN that employs a gating mechanism similar to that of LSTM units. In contrast to LSTMs, GRUs lack an output gate. This architecture addresses the vanishing gradient issue that may happen in ordinary recurrent neural networks. GRU is useful for textual data because it can handle continuous data, such as text, time series, audio, and images, *etc.* The architecture of GRU is shown in Fig. 4. GRU is more efficient because it only has two gates *i.e.,* update and reset gate. It determines what information to store in the hidden state and what to discard. This helps the network to overcome the problem of vanishing gradients in traditional RNNs which makes it hard to train longer sequences. Additionally, GRUs captures long-term relations in the data by maintaining a hidden state that can remember information from previous time steps. This is important for textual data, as an interpreted token or phrase often rely on the full sentence context provided by old tokens or phrases. In summary, GRU is useful for textual data because it can also overcome the issue of vanishing gradients, all of the above-mentioned features are important characteristics for processing and understanding natural language (*Chung et al., 2014*).

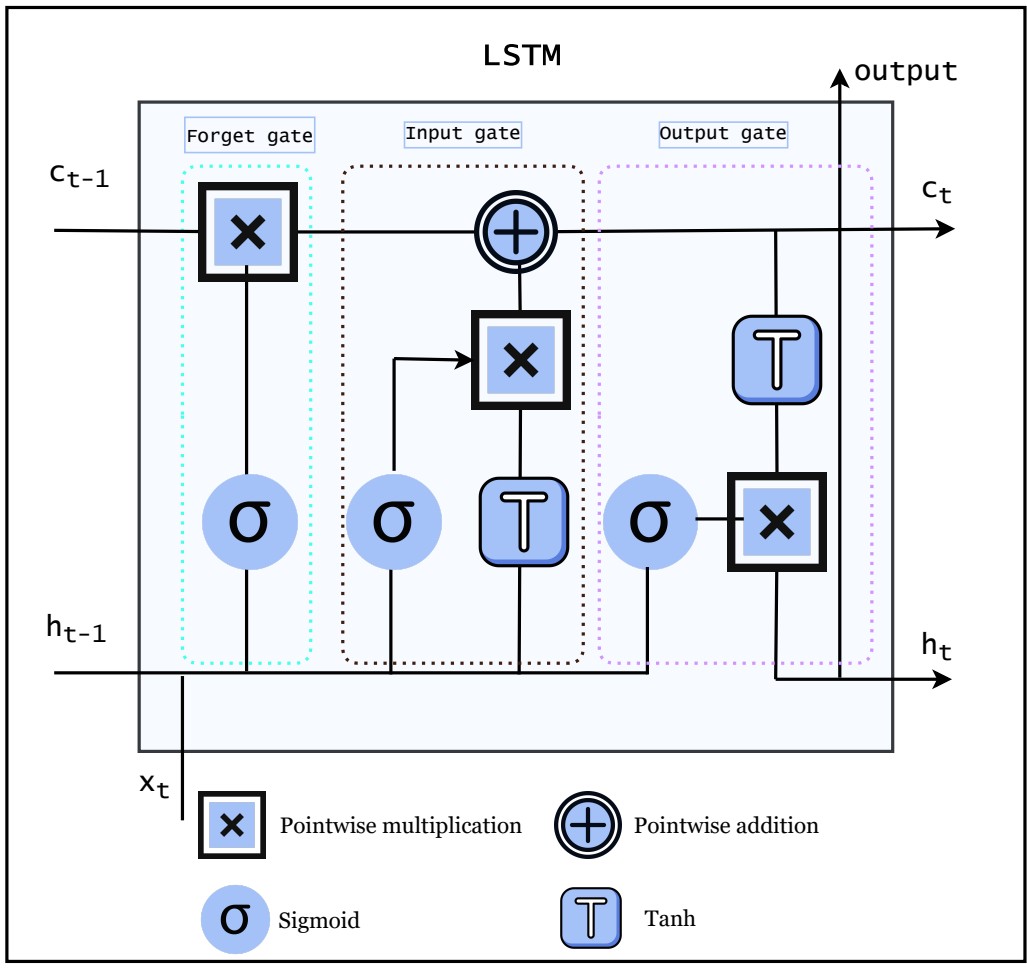

**Figure 3** **Architecture of LSTM** (*Imrana et al., 2021*).

### Bi-GRU

The recurrent neural network type known as a bi-GRU, or bidirectional gated recurrent unit, allows input to flow in both directions. This indicates that it can utilize both forward and backward sequence direction data. This architecture is especially helpful for simulating how words and phrases interact with one another in a sequence. It handles continuous data, such as text, by maintaining a hidden state that can remember information from previous time steps.

When a Bi-GRU is applied to textual data, it first converts the text into numerical representations, often through the use of embeddings. These embeddings are then fed into the forward and backward GRU layers, which process the data in their respective directions. The output from these layers is then combined and passed through a dense layer to produce the final output. The bidirectional nature of the Bi-GRU allows it to take into account both past and future context when processing the data, which can improve

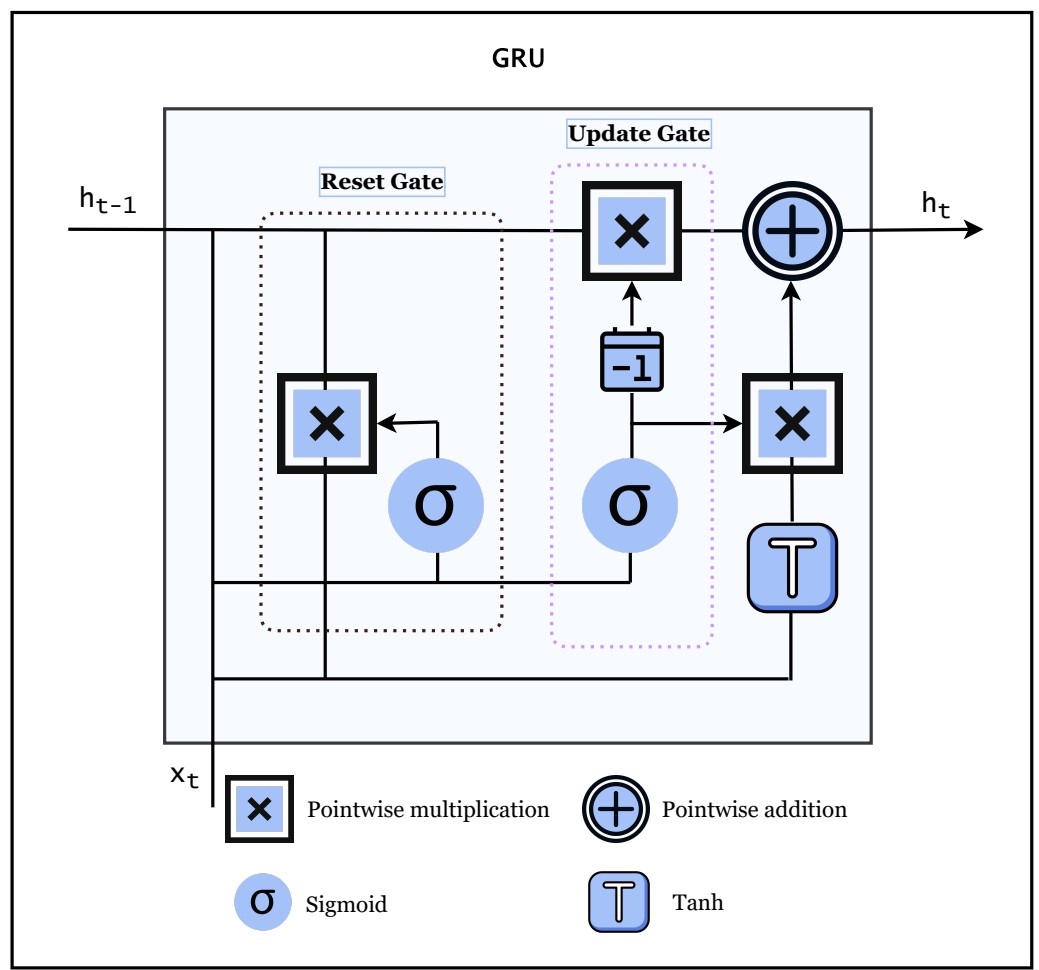

**Figure 4** Architecture of GRU (*Chung et al., 2014*).

its performance for tasks such as language modeling and sentiment analysis (*Yu, Wang & Jiang, 2021*). The architecture of BiGRU is shown in Fig. 5.

## Experimental setup
### Student feedback data set

The UIT-VSFC (*Van Nguyen et al., 2018*), a corpus of student evaluations from a Vietnamese university, is a set of data has been translated into English. This dataset contains 16,175 statements with various attitudes such as negative (0), positive (2), and neutral (1). Our dataset contains the following topics: curriculum or training program (1), lecturer (0), facilities (2), and other (3). The study collected feedback from students and teachers from 2013 to 2016, at the conclusion of each semester, using an automated survey system. The survey system used a 5-point Likert scale to evaluate the stated criteria and open-ended forms to gather more detailed feedback.

The dataset's legitimacy is determined by the inter-annotator agreement metric, which is a measure of how much agreement there is between different annotators when labeling

**Peer**J Computer Science

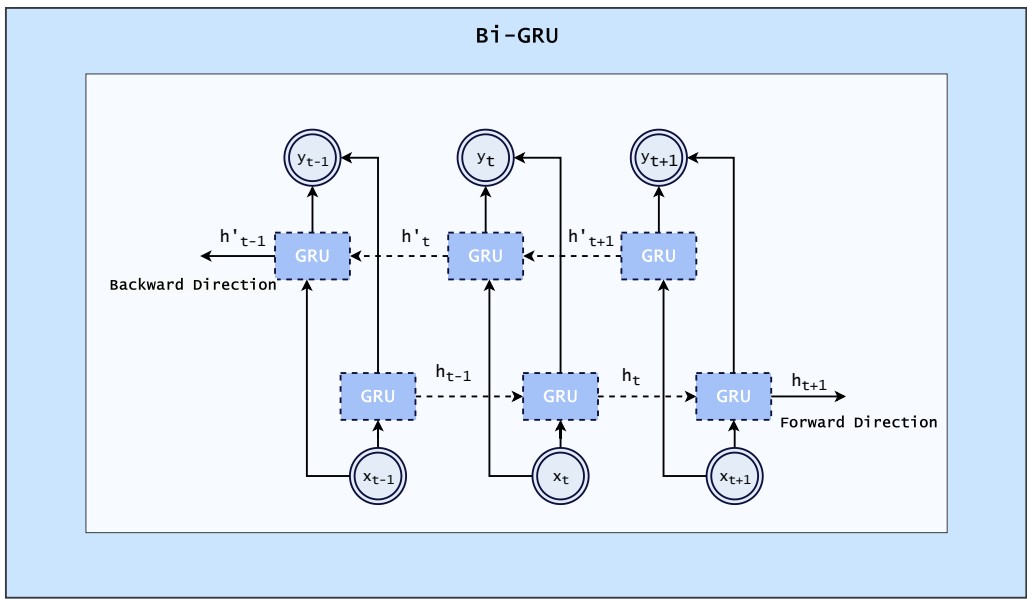

**Figure 5  Architecture of Bi-GRU (*Yu, Wang & Jiang, 2021*).**

the data. The inter-annotator agreement metric for this dataset is 91%, which means that 91% of the times, the multiple annotators who labeled the data agreed on the sentiments of the feedback. This high agreement rate is an indicator that the dataset is reliable, and can be deployed in solving sentiment analysis issues.

First, it is crucial to acknowledge that the data collection process was automated. This means that the data was collected using a system that operates without human intervention, which could limit the scope of the feedback gathered. This could be a limitation because the survey system may not have been able to capture certain nuances or personal experiences that are important for the study. Furthermore, the survey system used a 5-point Likert scale. The Likert scale is a commonly used tool in surveys to measure people's opinions and attitudes on a particular issue. However, its use could also be a limitation in the study, as it may make the data more prone to bias. Respondents may feel pressured to choose a response that conforms to social norms or expectations, rather than their true opinions. Additionally, the data collection period of 3 years could also be a limitation in the study. While this period may have allowed the researchers to gather a significant amount of data, it may not represent current opinions and trends. This is because opinions and attitudes can change over time, and the data collected three years ago may not accurately reflect the current state of affairs.

For experimental analysis the dataset was divided based on 80-20 rule, that is training (80%), validation (10%), and testing (10%) sets. It was ensured to avoid bias in dataset, for which a balanced samples across positive, negative, and neutral comments were selected.

Figure 6 depicts the class distribution for the sentiment of student feedback in the dataset. The class distribution for the topics of student feedback in the dataset is shown in Fig. 7.

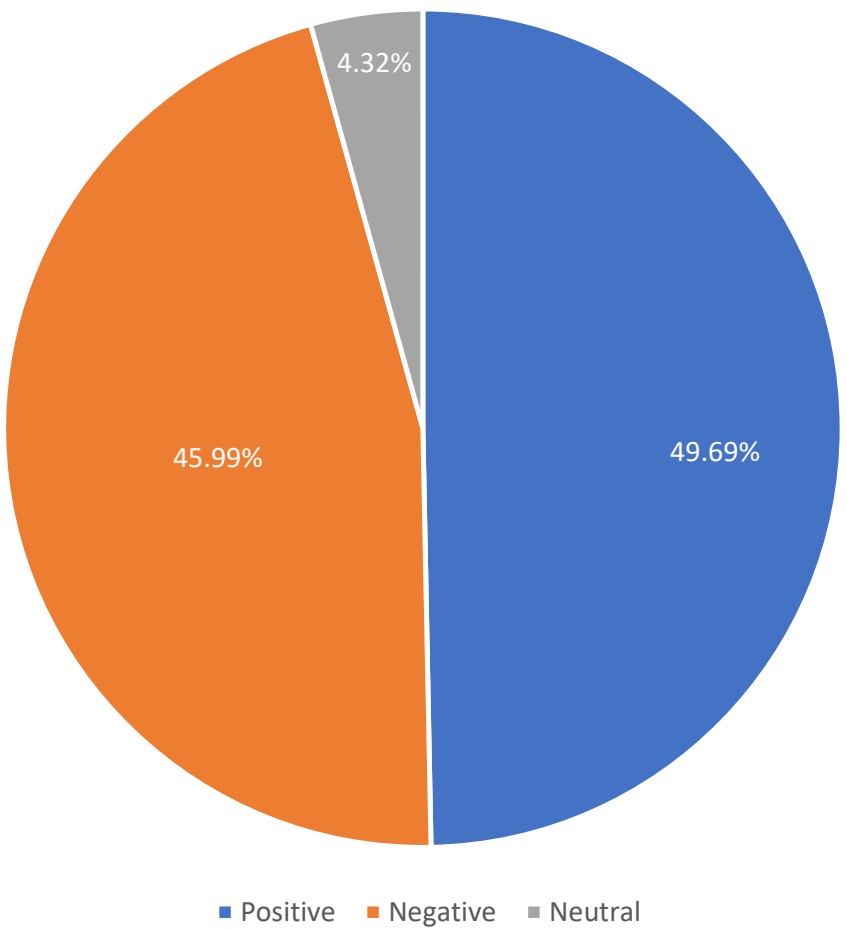

**Figure 6** **Total feedback for each sentiment.**

### Software and hardware

In the proposed research work, Python language in a Google Colab run-time environment-GPU using Windows 10 Pro for sentimental analysis of student feedback was used. The hardware used was Intel (R) Core (TM) i3-400 5U CPU @1.70 GHz Processor with the RAM of 4 GB.

### Hyper parameters

Hyper-parameters are deep learning model parameters that are specified before training begins and are not learnt during training. Hyper-parameters are used to manage the model's behavior and fine-tunes its performance. In deep learning, there are several types of hyper-parameters that can be adjusted to achieve accurate predictions, such as: learning rate, hidden layers, optimizer, batch size, activation functions, and many more. The hyper-parameters we used in our models are displayed in Table 1.

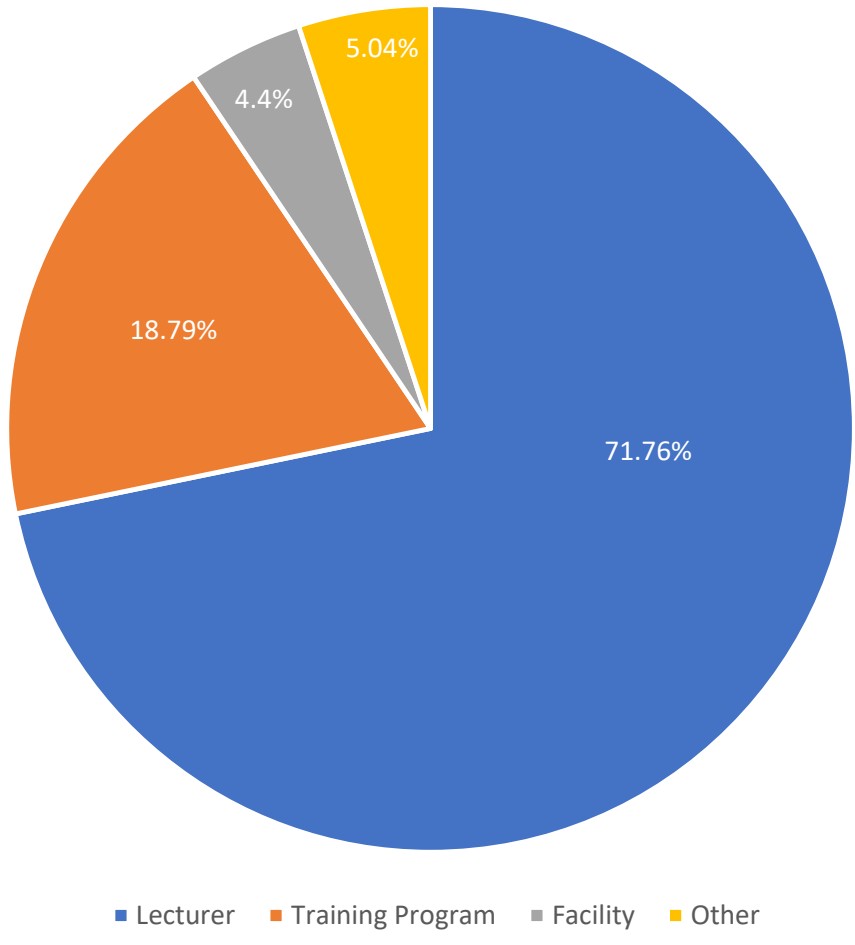

Total Feedback for each Sentiment Topics

- Lecturer  - Training Program  - Facility  - Other

**Figure 7** **Total feedback for each sentiment topics.**

## Model evaluation metrics

Evaluation metrics are vital to asses a model. There are various evaluation metrics that can be used according to the nature of working domain. For natural language processing domain, precision, recall, F1-score are used to evaluate the model. Further information of domain, precision, recall, F1-score are used to evaluate the model. Further information of these models are given below.

### *Precision*

It calculates the predicted true positive cases that are also correct in the dataset. It ranges between 0 and 1 and can be calculated as a macro-average or micro-average for a multi-class classification problem. The equation for precision for a single class "i" in a multi-class classification problem is:

$$\text{Precision} = \text{True Positives} / (\text{True Positives} + \text{False Positives}) \tag{1}$$

**Table 1   Hyper-parameters setting.**

| Hyper parameters | Values |
|---|---|
| Optimizer | Adam |
| Learning rate | 0.001/0.01 |
| Batch size | 256/128 |
| Epochs | 100 |
| Dropout ratio | 0.2 |
| Number of hidden layers | 9 |
| Number of neurons at classifying layer | 128 |
| Number of neurons at dense layers | 128, 64, 3, 4 |
| Activation function | tanh, relu, softmax |
| No of multi-heads | 10 |
| Fastext dimensions | 300 |
| Elmo dimensions | 1,024 |
| Early stopping | Yes |

### *Recall*

It calculates the true positive instances that were accurately identified by the trained It calculates the true positive instances that were accurately identified by the trained model. The equation for a single class ''i'' in a multi-class classification problem is:

$$\text{Recall} = \text{True Positives}/(\text{True Positives} + \text{False Negatives}) \tag{2}$$

It ranges between 0 and 1 and can be expressed as a percentage. A high recall value indicates that the model has a low false negative rate, meaning that it correctly identified most of the actual positive cases. It's important to note that, recall can be calculated for each class separately or it can be calculated as a macro-average or micro-average.

### *F1-score*

It merges the precision and recall to provide a concrete measurement of a model's performance. It ranges between 0 and 1 and can be calculated as a macro-average or micro-average for a multi-class classification problem. It's particularly useful when the classes are imbalanced. The equation for F1-score for a single class ''i'' in a multi-class classification problem is:

$$\text{F1-score} = 2*(\text{Precision} * \text{Recall})/(\text{Precision} + \text{Recall}) \tag{3}$$

To ensure the robustness and reliability, the results were computed based on 5-fold cross-validation.

## RESULTS

In this section, the numerical results of the proposed technique are discussed. The anticipated model finds the polarity of sentences *i.e.,* negative, positive, and neutral. The topics of a particular sentence is also found out *i.e.,* curriculum or training program, lecturer, facilities or others from the student feedback dataset.

**Table 2  Numerical results of sentiments using the proposed technique.**

| Classifier | Class | Precision | Recall | F1-score | Support |
|---|---|---|---|---|---|
| Bi-LSTM | Negative | 0.91 | 0.93 | 0.92 | 1,485 |
| | Neutral | 0.47 | 0.19 | 0.27 | 151 |
| | Positive | 0.91 | 0.94 | 0.93 | 1,599 |
| GRU | Negative | 0.90 | 0.96 | 0.93 | 1,485 |
| | Neutral | 0.54 | 0.22 | 0.31 | 151 |
| | Positive | 0.94 | 0.93 | 0.93 | 1,599 |
| Bi-GRU | Negative | 0.90 | 0.95 | 0.92 | 1,490 |
| | Neutral | 0.61 | 0.18 | 0.31 | 155 |
| | Positive | 0.92 | 0.94 | 0.93 | 1,645 |

**Table 3  Numerical results of topics using the proposed technique.**

| Classifier | Class | Precision | Recall | F1-score | Support |
|---|---|---|---|---|---|
| Bi-LSTM | Training | 0.01 | 0.93 | 0.91 | 2,292 |
| | Lecture Program | 0.67 | 0.67 | 0.67 | 616 |
| | Faculty | 0.91 | 0.80 | 0.84 | 162 |
| | Other | 0.50 | 0.30 | 0.38 | 165 |
| GRU | Training | 0.90 | 0.93 | 0.91 | 2,292 |
| | Lecture Program | 0.66 | 0.72 | 0.69 | 616 |
| | Faculty | 0.94 | 0.80 | 0.84 | 167 |
| | Other | 0.50 | 0.24 | 0.38 | 168 |
| Bi-GRU | Training | 0.90 | 0.91 | 0.90 | 2,082 |
| | Lecture Program | 0.60 | 0.73 | 0.66 | 750 |
| | Faculty | 0.93 | 0.75 | 0.83 | 151 |
| | Other | 0.58 | 0.17 | 0.26 | 128 |

Tables 2 and 3 show the empirical results on testing data by using the proposed model with a classifier (Bi-LSTM, GRU, and Bi-GRU) for sentiment and topic classification, respectively. The reports include four key metrics: precision, recall, F1-score, and support. Precision refers to the fraction of true positive predictions out of all positive predictions. Recall represents the fraction of true positive predictions out of all actual positive cases. The F1-score is the harmonic mean of precision and recall and provides a single number to represent the overall accuracy of the model. Support refers to the number of samples of the true response that lie in that class.

In these tables, one can easily determine the strengths and weaknesses of the model. For instance, if precision is high, but recall is low, it indicates that the model is good at identifying true positive cases, but not so good at finding all positive cases. On the other hand, if recall is high, but precision is low, it means that the model is detecting most of the positive cases, but some of them might be false positives.

### Comparative analysis of the proposed model with existing techniques

In the proposed work, three important issues in the field of natural language processing: polysemy, contextual meaning, and out-of-vocabulary terms were addressed. We employed three cutting-edge models to solve these problems: FastText, Elmo, and RoBERTa.

FastText is a text categorization and representation learning library that is intended to be quick and efficient. It handles out-of-vocabulary terms using sub-word information, making it appropriate for NLP jobs. Embeddings from Language Models (ELMO) is a contextualized word representation model that learns word contexts from large-scale text corpora. It captures the contextual meaning of words, making it ideally suited for NLP jobs requiring context awareness. A Robustly Optimized BERT Pre-training Approach (Roberta) is a transformer-based language model that has been refined using large-scale text corpora.

It outperforms BERT on a variety of NLP tasks, making it an effective model for NLP. The outputs of these three models were combined using ensemble stacking techniques and the results were passed through numerous levels of multi-head attention. This made it possible to identify the benefits of each model and use their problem-solving abilities to solve polysemy, contextual meaning, and vocabulary terms. We were able to generate predictions after passing the final output *via* classifier layers. In comparison to several baseline models and recently released models, our suggested model performed better. A comparison of our model to other models is presented in Table 4. In the proposed model, three classification techniques were used: Bi-LSTM, GRU, and Bi-GRU, as discussed in the 'Stacking: ensemble encoded features'. Bi-GRU classifier gives better results comparatively, therefore in the Table 4 only the Bi-GRU results are given in detail. The results for sentiments and topics were presented separately. Here, the average results of both categories (sentiments and topics) are given in Table 4 against the proposed model. The results demonstrate that, in terms of accuracy and other performance metrics, the proposed model performs better than existing models.

Figure 8 helps to further elaborate these results. This figure shows the comparative analysis of the proposed model with other techniques. Here the top four results of Table 4 are compared with our model. The results confirm that our proposed model is outperforming other models in all aspects such as accuracy, precision, recall and F1-score.

### Limitations and domain applicability of the proposed approach

The proposed approach is designed to student feedback, including domain-specific vocabulary, words, and phrases. In order to apply the proposed model to different domains (*e.g.*, sentiment analysis of the product reviews, healthcare feedback) may require further fine-tuning with domain specific data.

The proposed approach effectively utilizes the contextual information, but sarcasm and irony detection may impact the accuracy. It could be explored further to address this issue by integrating specialized techniques.

The proposed approach uses the multi-head attention mechanism, which requires more computational resources during training and inference. This could limit the model usability with limited resources environment.

**Table 4** Comparative analysis of the proposed model with existing state-of-the-art techniques.

| Models | Accuracy | Recall | Precision | F1-score |
|---|---|---|---|---|
| Fuzzy model (*Asghar et al., 2020*) | 89 | 97 | 87 | 90 |
| BERT-DCNN (*Jain et al., 2022*) | 86 | 81 | 87 | 86 |
| TF-IDF-LEX (*Nasim, Rajput & Haider, 2017*) | 93 | 90 | 90.80 | 92 |
| BICNN-RNN (*Song, Park & Shin, 2019*) | 81 | 90 | 85 | 82 |
| FT-CNN-BERT (*Luu, Nguyen & Nguyen, 2021*) | 86.88 | 84 | 82 | 88 |
| SMOTE-SVM (*Flores et al., 2018*) | 84 | 87 | 89 | 82 |
| BIGRU-LSTM- CNN-PHOW2V (*Nguyen, Nguyen & Nguyen, 2021*) | 79.40 | 85 | 88 | 62.69 |
| LSTM-DT (*Nguyen, Van Nguyen & Nguyen, 2018*) | 90.2 | 92 | 90 | 87 |
| RNTN-TB (*Socher et al., 2013*) | 80 | 77 | 85 | 78 |
| ATSA (*Peng, Xiao & Yuan, 2022*) | 86 | 89 | 87 | 90 |
| RU-BiLSTM (*Chandio et al., 2022*) | 67 | 75 | 65 | 72 |
| CHL-PRAE (*Fu et al., 2017*) | 83.9 | 80 | 79 | 87 |
| Proposed model | 95 | 97 | 95 | 96 |

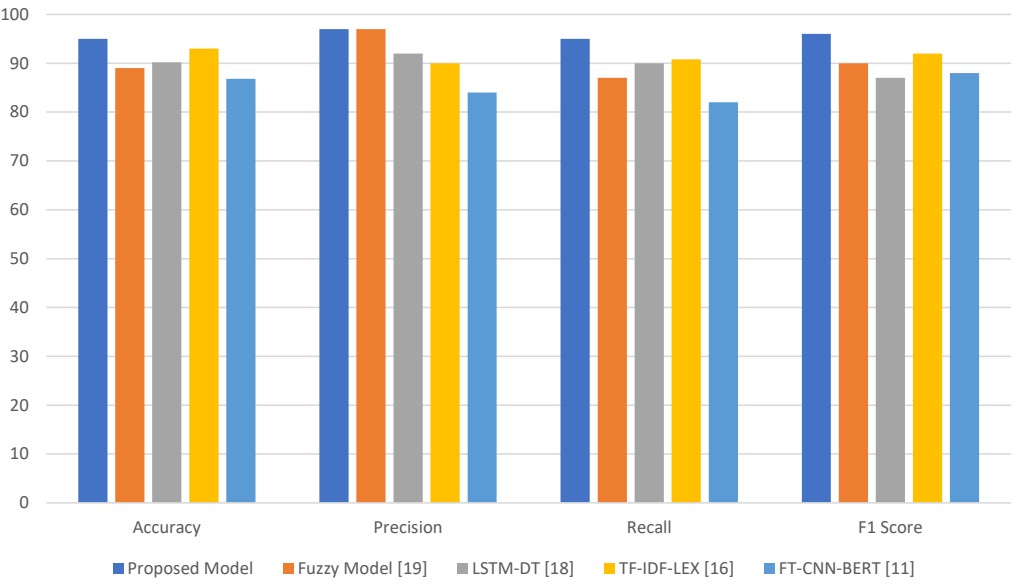

**Figure 8** Comparative analysis of the proposed model with other techniques.

# CONCLUSIONS

Accurate sentiment or emotion classification of student feedback is potent for the accurate sentiment or emotion classification of student feedback is potent for the success of any educational institution. It is crucial to understand the emotions, thoughts, and opinions of students in regards to the faculty, curriculum, training programs, facilities, and more. The proposed model addresses issues with word morphology and polysemy by utilizing

techniques such as lemmatization and stemming to treat inflected forms of words as a single entity, while also considering their semantics. This allows for phrases made up of words with similar meanings to be grouped together, making the text easier to interpret and leading to more accurate sentiment analysis results. Additionally, the proposed model effectively captures the contextualization and local context of student feedback by utilizing techniques such as attention mechanisms, Fastext, RoBERTa and ELMO to extract features of phrases and sequences of textual data. This allows for a more thorough understanding of the context in which the feedback was given, leading to more accurate sentiment analysis results. By using the proposed multi-layer hybrid model, which combines text embedding techniques, multi-head attention mechanism, and deep learning-based models such as Bi-LSTM, GRU, and BiGRU, achieved more accurate results in the classification of textual data of student feedback as compared to the state-of-the-art methods. Hence, in light of the above points we can suggest our model to other textual data too. However, further evaluations can be performed to establish this claim.

Sentiment analysis is an evolving field, and in the future, we can expect to see and work on advancements in natural language understanding, multilingual sentiment analysis, incorporating context, real-time sentiment analysis, and sentiment analysis on multimedia data. Moreover, there are several areas that can be explored to improve sentiment analysis in the future. These include incorporating unstructured data such as images, audio, and video, incorporating contextual information, developing models that can handle low-resource languages, incorporating user-generated data and incorporating domain-specific knowledge. In order to apply the proposed model to different domains (*e.g.*, sentiment analysis of the product reviews, healthcare feedback) may require further fine-tuning with domain specific data. Sarcasm and irony detection could be explored further by integrating specialized techniques. These advancements will lead to more accurate results, making sentiment analysis more accessible to a larger audience and more useful for businesses and organizations.

### Funding
The authors received no funding for this work.

### Competing Interests
Joanna Rosak-Szyrocka is an Academic Editor for PeerJ.

### Author Contributions
- Shanza Zafar Malik conceived and designed the experiments, performed the experiments, performed the computation work, prepared figures and/or tables, and approved the final draft.
- Khalid Iqbal performed the experiments, performed the computation work, authored or reviewed drafts of the article, and approved the final draft.

- Muhammad Sharif performed the experiments, analyzed the data, performed the computation work, prepared figures and/or tables, and approved the final draft.
- Yaser Ali Shah conceived and designed the experiments, authored or reviewed drafts of the article, and approved the final draft.
- Amaad Khalil analyzed the data, authored or reviewed drafts of the article, and approved the final draft.
- M. Abeer Irfan analyzed the data, authored or reviewed drafts of the article, and approved the final draft.
- Joanna Rosak-Szyrocka performed the computation work, authored or reviewed drafts of the article, and approved the final draft.

## Data Availability

The data is available at figshare: Irfan, Muhammad Abeer (2024). Raw Data.zip. figshare. Dataset. https://doi.org/10.6084/m9.figshare.25427797.v1.

## Supplemental Information

Supplemental information for this article can be found online at http://dx.doi.org/10.7717/peerj-cs.2283#supplemental-information.

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
