# Peer review of "Attention-aware with stacked embedding for sentiment analysis of student feedback through deep learning techniques"

_PeerJ Computer Science, doi:10.7717/peerj-cs.2283_

## Round 0.1 · original submission · Major Revisions

Based on the reviewer comments, the manuscript must be revised before acceptance.

Reviewer 1 ·

Basic reporting

The manuscript entitled “Attention-aware with stacked embedding for sentiment analysis of student feedback through deep learning techniques” has been investigated in detail. This research explores the challenge of automatic polarity prediction in Natural Language Processing (NLP), particularly in the context of student feedback. The study proposes an innovative hybrid model that combines ensemble learning, text embedding, a multi-head attention mechanism, and deep learning classifiers to analyze sentiments in student comments. There are some points that need further clarification and improvement:
1) The paper lacks a clear explanation of how the proposed hybrid model was constructed and the specific role of each component (text embedding, multi-head attention, and deep learning classifiers) in the ensemble approach. More details on the architecture and implementation are needed.
2) While the paper claims high accuracy, recall, precision, and F1-score, there is insufficient discussion on how these metrics were calculated, the datasets used for training and testing, and whether there was any bias in the data.
3) Although the paper states that the proposed model outperforms existing techniques, there is little discussion on what those techniques are and how the comparison was conducted. More rigorous benchmarking and analysis are needed.

Experimental design

The paper lacks information on the availability of the dataset and the code used for the experiments, making it difficult to replicate the study and validate the results independently.

The proposed model's performance is impressive, but there is no discussion on how it generalizes to different types of text data beyond student feedback. The model may need further validation with other datasets to establish its robustness.

Validity of the findings

The paper's novelty and contribution to the field of sentiment analysis are not well-articulated. The proposed hybrid model needs to be compared against other similar models to establish its significance.

The paper does not discuss the limitations of the proposed approach, including potential challenges in applying the model to different domains or with different types of data.

Additional comments

The manuscript entitled “Attention-aware with stacked embedding for sentiment analysis of student feedback through deep learning techniques” has been investigated in detail. This research explores the challenge of automatic polarity prediction in Natural Language Processing (NLP), particularly in the context of student feedback. The study proposes an innovative hybrid model that combines ensemble learning, text embedding, a multi-head attention mechanism, and deep learning classifiers to analyze sentiments in student comments. There are some points that need further clarification and improvement:
1) The paper lacks a clear explanation of how the proposed hybrid model was constructed and the specific role of each component (text embedding, multi-head attention, and deep learning classifiers) in the ensemble approach. More details on the architecture and implementation are needed.
2) While the paper claims high accuracy, recall, precision, and F1-score, there is insufficient discussion on how these metrics were calculated, the datasets used for training and testing, and whether there was any bias in the data.
3) Although the paper states that the proposed model outperforms existing techniques, there is little discussion on what those techniques are and how the comparison was conducted. More rigorous benchmarking and analysis are needed.
4) The paper lacks information on the availability of the dataset and the code used for the experiments, making it difficult to replicate the study and validate the results independently.
5) The proposed model's performance is impressive, but there is no discussion on how it generalizes to different types of text data beyond student feedback. The model may need further validation with other datasets to establish its robustness.
6) The paper's novelty and contribution to the field of sentiment analysis are not well-articulated. The proposed hybrid model needs to be compared against other similar models to establish its significance.
7) The paper does not discuss the limitations of the proposed approach, including potential challenges in applying the model to different domains or with different types of data.
The paper presents an innovative hybrid model for sentiment analysis using ensemble learning, attention mechanisms, and deep learning classifiers. While the results are promising, the study lacks sufficient methodological details, a thorough evaluation process, and a clear articulation of the paper's contributions to the field. The authors should address these issues and provide more evidence of the model's robustness and generalizability.

Reviewer 2 ·

Basic reporting

The paper proposes an innovative hybrid model but does not provide sufficient evidence of its novelty. The use of ensemble learning, multi-head attention, and deep learning classifiers is well-established in the field, and the paper does not demonstrate a significant departure from existing approaches.

While the paper claims the model outperforms state-of-the-art techniques, it lacks extensive benchmarking against other contemporary models. A more comprehensive comparison with existing methods across different datasets would strengthen the claims.

Experimental design

The focus on student comments may limit the generalizability of the model to other domains or types of data. The paper should discuss how the model can be adapted or applied to other contexts.

The study primarily focuses on student responses, which may not be representative of other user comments in broader contexts such as Twitter and Facebook. More diverse data sources could strengthen the model's applicability.

Validity of the findings

The paper emphasizes performance metrics such as accuracy, recall, precision, and F1-score, but it does not provide a detailed analysis of the model's limitations or potential biases. Discussing these aspects would add more depth to the evaluation.

The paper should provide more clarity on the ensemble learning, text embedding, and attention mechanism used in the hybrid model. A deeper discussion on these components' implementation and how they work together would help the reader understand the model better.

Additional comments

Paper Title: Attention-aware with stacked embedding for sentiment analysis of student feedback through deep learning techniques
Comments:
1) The paper proposes an innovative hybrid model but does not provide sufficient evidence of its novelty. The use of ensemble learning, multi-head attention, and deep learning classifiers is well-established in the field, and the paper does not demonstrate a significant departure from existing approaches.
2) While the paper claims the model outperforms state-of-the-art techniques, it lacks extensive benchmarking against other contemporary models. A more comprehensive comparison with existing methods across different datasets would strengthen the claims.
3) The paper emphasizes performance metrics such as accuracy, recall, precision, and F1-score, but it does not provide a detailed analysis of the model's limitations or potential biases. Discussing these aspects would add more depth to the evaluation.
4) The paper should provide more clarity on the ensemble learning, text embedding, and attention mechanism used in the hybrid model. A deeper discussion on these components' implementation and how they work together would help the reader understand the model better.
5) The focus on student comments may limit the generalizability of the model to other domains or types of data. The paper should discuss how the model can be adapted or applied to other contexts.
6) The study primarily focuses on student responses, which may not be representative of other user comments in broader contexts such as Twitter and Facebook. More diverse data sources could strengthen the model's applicability.
The paper presents an interesting approach to automatic polarity prediction using a hybrid model. However, it needs to address the concerns mentioned above regarding novelty, benchmarking, and methodological transparency. Additionally, the paper should consider the model's applicability to other domains and provide more diversity in data sources. Overall, the research shows potential, but significant revisions are needed for clarity and rigor.

---

## Round 0.2 · accepted · Accept

Based on the reviewers comments, the manuscript can be accepted.

Reviewer 1 ·

Basic reporting

My comments have been addressed. It is acceptable in the present form.

Experimental design

My comments have been addressed. It is acceptable in the present form.

Validity of the findings

My comments have been addressed. It is acceptable in the present form.

Reviewer 2 ·

Basic reporting

The paper may be accepted for publication in its current form.

Experimental design

The paper may be accepted for publication in its current form.

Validity of the findings

The paper may be accepted for publication in its current form.